# Thermal Modification of Spruce and Maple Wood for Special Wood Products

**DOI:** 10.3390/polym14142813

**Published:** 2022-07-10

**Authors:** Anna Danihelová, Zuzana Vidholdová, Tomáš Gergeľ, Lucia Spišiaková Kružlicová, Michal Pástor

**Affiliations:** 1Department of Fire Protection, Faculty of Wood Sciences and Technology, Technical University in Zvolen, T. G. Masaryka 24, 960 01 Zvolen, Slovakia; 2Department of Wood Technology, Faculty of Wood Sciences and Technology, Technical University in Zvolen, TU in Zvolen, Masaryka 24, 960 53 Zvolen, Slovakia; zuzana.vidholdova@tuzvo.sk; 3National Forest Centre, Forest Research Institute, T. G. Masaryka 22, 960 01 Zvolen, Slovakia; tomas.gergel@nlcsk.org (T.G.); michal.pastor@nlcsk.org (M.P.); 4Department of Furniture Design and Interior, Faculty of Wood Sciences and Technology, Technical University in Zvolen, TU in Zvolen, Masaryka 24, 960 53 Zvolen, Slovakia; kruzlicovalucia@gmail.com

**Keywords:** thermal modification, spruce and maple wood, physical and acoustical characteristics, timbre of sound, musical instruments

## Abstract

This article presents a proposal of thermal modification of Norway spruce and sycamore maple for special wood products, mainly for musical instruments. Selected physical and acoustical characteristics (PACHs), including the density (*ρ*), dynamic modulus of elasticity along the wood grain (*E*_L_), specific modulus (*E*_sp_), speed of sound along the wood grain (*c*_L_), resonant frequency (*f*_r_) and acoustic constant (*A*), logarithmic decrement (*ϑ*), loss coefficient (*η*), acoustic conversion efficiency (*ACE*), sound quality factor (*Q*), and the timbre of sound, were evaluated. These two wood species were chosen regarding their use in the production or repair of musical instruments. For the thermal modification, a similar process to the ThermoWood process was chosen. Thermal modification was performed at the temperatures 135 °C, 160 °C and 185 °C. The resonant dynamic method was used to obtain the PACHs. Fast Fourier transform (FFT) was used to analyze the sound produced. The changes in the observed wood properties depended on the treatment temperature. Based on our results of all properties, the different temperature modified wood could find uses in the making of musical instruments or where the specific values of these wood characteristics are required. The mild thermal modification resulted in a decrease in mass, density, and increased speed of sound and dynamic modulus of elasticity at all temperatures of modification. The thermally modified wood showed higher sound radiation and lower loss coefficients than unmodified wood. The modification also influenced the timbre of sound of both wood species.

## 1. Introduction

The modification of wood properties using heat has been used for many years in the manufacture of musical instruments, where the technique is used to replicate the highly desirable tones and inherent stability of aged guitars and violins. String musical instruments have come a long way since their inception. The development has reached a stage where it is possible to move materially and structurally only within certain limits (e.g., due to prestressing, the volume and mass of the board can be reduced or the center of gravity of the board can be shifted). It is, therefore, desirable to focus on the material from which the musical instrument is made.

Musical instrument manufacturers have been using wood that is naturally dried for more than 5 years. The aging of wood can be defined as slow chemical reactions in wood that occur over time under variations in climatic conditions [1]. These chemical changes, as well as the changes in the microstructure of wood, are the cause of the changes in the physical, acoustical, and mechanical properties of wood [2,3]. Long-term natural drying reduces the hygroscopicity of wood. The reduced hygroscopicity improves the dimensional stability of wood, as well as stabilizing its mechanical and acoustic properties. The speed of sound increases and the internal damping decreases with a reduction in moisture content; therefore, the musical instrument often sounds noticeably brighter [4,5].

Obtaining long-term aging wood is becoming increasingly difficult; therefore, manufacturers of musical instruments are becoming increasingly interested in various methods that allow the simulation of the effects of long-term natural drying wood. The aim of wood modification is to achieve the required properties of wood for musical instruments. Mechanical modification (densification) of wood, thermal modification, or their combination, appear to be suitable modifications of wood for musical instruments [6,7,8,9,10,11,12]. Similarly, the biological modification of wood with wood-staining fungi or wood-destroying fungi can be used to modify the relevant properties [13,14,15]. These modifications of wood improve its properties, producing material that when disposed at the end of the product life cycle, does not present an environmental hazard any greater than unmodified wood [16]. Heat treatment as a wood modification process is based on chemical degradation of wood by heat transfer [17]. In the heat treatment process, wood is heated to temperatures ranging from 180 °C to 260 °C, where lower temperatures cause minor changes in wood components and higher temperatures cause severe changes [16]. Thermal modification causes the number of hydroxyl groups of cellulose and hemicelluloses to decrease, thereby resulting in the decrease in the absorption of water [18,19]. Thermally modified wood is dimensionally more stable, which is beneficial to wind instruments (due to changes in relative humidity, the instrument does not lose tuning. The increase in dimensional stability [20] reduces the formation of cracks and improves the efficiency of paint coating systems. Higher dimensional stability can give greater tuning stability, which means that instruments will need to be tuned less [7]. Thermally modifying the wood also changes other important wood properties, such as biological durability, hardness, and UV-stability [18,21,22]. This is very important in wooden organ pipe making. The thermal modification also induces a darker coloration of the wood. The darkness intensity is dependent upon the treatment time and temperature [23]. Different wood species will turn a different darker shade during the process, due to their natural characteristics. The darker shade of wood color is welcome for some kinds of musical instruments (marimba, xylophone, guitar, etc.); however, this is provided that the wood has the required physical and acoustic properties in addition to color [6,24,25].

The aim of thermal modification of wood for musical instruments making is to maximize sonic benefits and minimize degradation (in guitar-making, this process is also known as “thermo curing”, “wood torrefaction” or “roasting” [25]). Guitar manufacturers have begun using acoustic sound boards and electric guitar fretboards that are thermally treated to help prevent the warping and cracking that often occurs. As a secondary benefit, the acoustic guitars (also violins) tend to sound similar to well-broken-in aged instruments much sooner than instruments made from thermally unmodified wood [24].

Based on the above-mentioned properties of long-term stored wood and thermally modified wood, it can be stated that relatively mild thermal treatment will accelerate changes in the structure, and thus also in the physical and acoustic properties of the wood [8,26]. The quality of a string musical instrument strongly depends on that of its soundboard. It is generally known that a high acoustic constant, low density, and a large degree of anisotropy are required for an excellent soundboard [27,28]. The resonant wood (for string musical instruments) should have a low density *ρ* (around 430 kg∙m^−3^), a low logarithmic damping decrement ϑ (around 0.022) and a high acoustic constant *A* (around 13 m^4^·kg^−1^∙s^−1^). On the other hand, maple wood should have a high density (around 600 kg∙m^−3^), a low acoustic constant *A* (around 6.5 m^4^·kg^−1^∙s^−1^) and logarithmic damping decrement ϑ (around 0.05). These relevant properties of used wood are important because flexible wood tends to shape well and projects the highest quality of sound [27,29,30,31].

The main objective of this study was to propose the appropriate regime of thermally modifying spruce and maple wood (for producing musical instruments) to improve their physical and acoustical characteristics (density, dynamic modulus of elasticity, acoustic constant, speed of sound, logarithmic damping decrement, etc.). The second objective was to determine the effect of thermal modification on the sound.

## 2. Material and Methods

Wood specimens were prepared from Norway spruce (*Picea abies* (L.) Karst.) and Sycamore maple (*Acer pseudoplatanus*, L.) growing in Slovakia. The specimens were prepared after natural drying the wood outdoors. The specimens from spruce and maple wood were sawn radially into the shape of a bar with dimensions (10 × 10 × 400) mm. The specimens from spruce wood had an annual ring width, which is typical for the bass to discant zone of pianos [32]. The specimens made from maple wood included curly maple and maple with straight grains. The specimens were divided into three sets for thermal modification at temperatures 135 °C, 160 °C and 185 °C. Each set consisted of 30 specimens.

The experiment was divided into three parts. In the first part of the experiment, the specimens were conditioned to 8% moisture at (20 ± 2) °C and (45 ± 5)% relative humidity to avoid splitting during thermal modification. After conditioning, the physical and acoustical characteristics (density *ρ*, modulus of elasticity along the grain *E*_L_, acoustic constant *A*, speed of sound *c*_L_, logarithmic damping decrement *ϑ*, loss coefficient *η*, acoustic conversion efficiency (*ACE*) and sound quality factor *Q*), the dimensions, volume, mass and moisture content of natural spruce and maple wood were determined. The moisture content of the samples after conditioning was in the intervals (8.0 ± 0.2)%.

The second part of the experiment was thermally modifying the specimens. The thermal modification used in this experiment used the ThermoWood process [33]. This method enables the modifying of the wood at lower temperatures (smaller decomposition/depolymerization of the construction polymers of wood) [34]. This is a “dry technology” wood that is modified in air, and the oil is not used to increase the interaction temperature, as there is no need to create an environment of steam typical for the PlatoWood process [35].

The thermal modification of wood consisted of the following three steps (Figure 1): high temperature drying (at 100 °C for 90 min and then the steadily increasing to 130 °C for 2.5 h); heat treatment at t = 135 °C, 160 °C and 185 °C for a 50 min period, cooling and moisture conditioning. The wood was cooled down in a controlled way so that its temperature reached 80 °C. At the end of the thermal modification, the kiln was turned off and the specimens were kept inside. The specimens were allowed to cool naturally until their temperature reached 20 °C.

After thermal modification, all specimens underwent conditioning under the same conditions as before the thermal modification to stabilize wood moisture content. After the stabilization of moisture content of wood (i.e., after reaching the equilibrium humidity of the wood) the physical and acoustical characteristics (density *ρ*, modulus of elasticity along the grain *E*_L_, acoustic constant *A*, speed of sound *c*_L_, logarithmic damping decrement *ϑ*, loss coefficient *η*, acoustic conversion efficiency (*ACE*), sound quality factor *Q*), the dimensions, volume, mass, and moisture content) as well as dimensions, volume and mass of specimens were measured again.

The moisture content of spruce and maple species were 7.0 ± 0.2% and 5.8 ± 0.2%, respectively, because of thermal modification at 135 °C and 185 °C temperatures. After modifying at 160 °C, the moisture contents (MC) of the spruce and maple wood specimens were different. MC of the spruce wood was 6.4 ± 0.2%, but MC of maple wood reached only 5.8 ± 0.2%.

For obtaining the relevant physical and acoustical characteristics (PACH) before and after modification, the resonant dynamic method was used. The principle of this method searches for the resonant frequency *f*r of a vibrating body (usually of bar shape), for which the dynamic modulus of elasticity along the grain *E*_L_ can be calculated. The measuring device MEARFA (Figure 2) was used for the measurements of relevant characteristics.

The device consists of a computer (1); generator of sinusoidal signal (in the range of acoustic frequencies) and response detector (2); exciter-loudspeaker (3); test specimen (4); electromagnetic detector (5); high-pass filter (6); preamplifier (7); digital gauger, Mitutoyo (8) and electronic scale (9). The fundamental resonance frequency *f*r, as well as frequencies *f*_1_ and *f*_2_, were determined. These were stored in the computer for further processing and evaluation using the computer software. The physical and acoustical characteristics (density *ρ*, modulus of elasticity along the grain *E*_L_, acoustic constant *A*, speed of sound *c*_L_ and logarithmic damping decrement *ϑ*) were calculated according to the Equations (1)–(4).

The dynamic modulus of elasticity along the wood grain *E*_L_ (Pa) of the specimen in the shape of bar was calculated [4,27] using the following equation:(1)$$E_{L}=4.l.f_{r}^{2}.\rho$$
where *ℓ* (m) is the length of the specimen, *ρ* (kg·m^−3^) is the density and *f*r (Hz) is the resonant frequency. The longitudinal specific modulus of elasticity *E*_sp_ (m^2^·s^−2^) was calculated with the equation
(2)$$E_{\mathrm{sp}}=\frac{E_{L}}{\rho }$$

The acoustic constant (also well known as sound radiation coefficient—*R*) *A* (m^4^·kg^−1^·s^−1^) was calculated [27,29] using the equation
(3)$$A=\sqrt{\frac{E_{L}}{\rho^{3}}}=\frac{c_{L}}{\rho }$$
where *E*_L_ (Pa) is the modulus of elasticity along the wood grain; *c* (m∙s^−1^) is the speed of sound. For calculating the speed of sound in wood [29], the following equation was used:(4)$$c_{L}=\sqrt{\frac{E_{L}}{\rho }}$$
where *E*_L_ (Pa) is the modulus of elasticity along the wood grain; *ρ* (kg∙m^−3^) is the density. The logarithmic damping decrement (expresses the absorption of acoustic energy by the material) was calculated [27] by the equation
(5)$$\vartheta =\frac{\pi }{\sqrt{3}}.\frac{f_{2}-f_{1}}{f_{r}}$$
where *f*_1_ and *f*_2_ (Hz) are frequencies at which the amplitude of the vibrations is half the maximum amplitude at resonant frequency *f*r.

Given the importance of vibration damping in musical instruments, the internal friction is also relevant. This is an intrinsic material property, unlike other loss mechanisms, such as the radiation of acoustic energy. An indicator of internal friction is the loss coefficient *η*. It was measured through the logarithmic decrement and calculated with the equation
(6)$$\eta =\frac{\vartheta }{\pi }$$

Combining *A* (m^4^·kg^−1^·s^−1^) and *η*, the acoustic conversion efficiency (*ACE*) (m^4^·kg^−1^·s^−1^) was calculated according to the equation
(7)$$ACE=\frac{A}{\eta }$$

The *ACE* is useful to show group effects, e.g., the effect of internal friction and sound radiation together [37,38]. The *η* is related to the sound quality factor *Q*, which represents the mechanical gain of a structure at a resonant frequency. The *Q* is a descriptor of the wood sound quality because it provides information about the resonance sharpness. The *Q* was calculated using the equation
(8)$$Q=\frac{1}{\eta }$$

The characteristic acoustic impedance *z* (MPa∙s∙m^−1^) represents the resistance of wood to sound wave propagation. It is important for musical instruments because it describes vibration transfer. A high value of characteristic acoustic impedance presents a high rate of reflection of sound wave by the medium with which it travels in. At the boundary between the media of different acoustic impedances, some of the wave energy is reflected and some is transmitted. Two materials with a large difference in values of characteristic acoustic impedance give much larger reflections as the transmissions. The characteristic acoustic impedance *z* was calculated using the equation
(9)$$z=\rho .c_{L}$$

The physical and acoustical characteristics were evaluated using Statistica software (Statsoft Inc., Version 7, Prague, Czech Republic), the two-way analysis of variance (ANOVA) as well as correlation analyses with the coefficient of determination (R2) and Duncan’s multiple range test to determine the significance of the variation at a 0.05 significance level (α).

The third part of experiment was fast Fourier transform (FFT), which is a digital implementation of the Fourier transform. The musical instruments in fact produce sound as a result of the vibration of strings, bars, or plates. Vibration causes a periodic variation in air pressure that is heard as a sound and these sound waves can be analyzed using Fourier series. The sound produced by musical instruments of the same kind differs, because they have different timbre sounds. The reason for this is that the energy in each of the harmonics is different. The Fourier analysis provides mathematical evidence of the dissimilarities of musical timbre sound.

The output of the frequency analysis is the frequency response, showing the sound intensity levels of the individual frequency components of the signal. The impulse excitation technique was used (described by [39]) to excite the bar vibrations. For our experiment, the experimental device (shown in Figure 3) was designed to simulate the conditions resembling those in which a musician would play a xylophone. The important part of device is the mechanical system of impulse excitation. This system ensures the same conditions of excitation for all specimens. The mechanical system consists of a pendulum that hangs on a metal rod embedded in the plates of plexiglass. A pendulum consists of a nylon cord (*ℓ* = 0.3 m) and a metal ball (*d* =14 mm, *m* = 12 g). The pendulum was set in motion to trigger a vibration in the wood bar by hitting the end with the metal ball. The system is designed to trigger a vibration in the wood bar by hitting it at the bottom of the bar at a distance of 10 cm from its end.

The test specimen was fastened in the node line of the 4th (2, 0) mode of vibration. The position of the nodes was determined as the excitation of the 4th mode of vibration before FFT analysis. The vibration frequency of the supporting system was low (below 10 Hz). The microphone was placed on a separate holder above the specimen at a distance of 25 mm from the other bar end to measure the acoustic pressure that radiates at impact. The sounds were produced and recorded in the laboratory in a free field.

The FFT analysis of sound before and after thermal modification of spruce and maple wood was performed using Adobe Audition program 1.5.

## 3. Results and Discussion

The mean values of the physical and acoustical characteristics (density *ρ*, dynamic modulus of elasticity *E*_L_, acoustic constant *A*, speed of sound *c*_L_, resonant frequency *f*r, logarithmic damping decrement *ϑ*, loss coefficient *η*, acoustic conversion efficiency (*ACE*) and sound quality factor *Q*) of spruce and maple specimens before thermal modification and after thermal modification at the temperatures 135 °C, 160 °C and 185 °C are given in Table 1.

### 3.1. Effect of Thermal Modification on Density

Table 1 shows that thermal treatment caused a decrease in spruce wood (SW) density at each temperature used in our experiments. The largest density decrease (on average around 2.8%) was recorded at 160 °C. The mean values of maple wood (MW) density reduction varied between 0.8 and 1.3%, in which the highest reduction was found after the treatment at the temperature of 135 °C. According to the ANOVA test, there is no statistically significant difference between the density of thermal modified and unmodified wood (F = 2.28, *p* > 0.05). Positive correlation between the temperature of modification and density was observed for both wood species (R^2^ = 0.83 for SW; R^2^ = 0.63 for MW).

The density reduction was mainly due to a reduction in mass (SW varied from 2.1% to 3.6%; MW from 2.5% to 4.6%) and a decrease in volume (SW from 1.8% to 2.9%; MW 1.2% to 3.3%), which was lower than the corresponding mass reduction. Our results showed that the mass loss of both wood species increased with temperature, which agrees with earlier results for spruce [40] and maple wood [35]. The mass loss values were similar to the results found in other studies of thermally modified softwoods and hardwoods [41,42]. The extent depends upon temperature and treatment time [43,44,45].

Thermal modification at lower temperatures led to a lower mass reduction associated mainly with the loss of volatiles and bound water. Higher mass loss was observed when the samples were treated at temperatures above 150 °C, which is the effect of the partial decomposition of hemicellulose and cellulose, but also other changes in the chemical structure of the thermally modified wood [35,46]. The intensity of thermic degradation (mass loss) maple wood was higher than spruce wood. It can be explained through the differences in chemical composition of hardwoods and softwoods because hardwoods contain slightly more hemicelluloses (25%) compared to softwoods (20%), which are the least resistant to thermal degradation [47,48].

### 3.2. Effect of Thermal Modification on Modulus of Elasticity

Mean values of modulus of elasticity (MOE) of spruce wood after modification increased by 5.1% (at 135 °C), by 4.1% (at 160 °C), and only by 1.2% after modification at 185 °C. This is in agreement with Millett and Gerhards [48]. Navickas et al. [49] reported slightly higher MOE of spruce wood (by 0.4%) than it was before heating (at 190 °C). Likewise, Kubojima et al. [50] found that heat treatment at 160 °C resulted in increased dynamic MOE. The modulus of elasticity maple wood increased by 6.0%, 6.8% and 3.0% (at 135 °C, 160 °C and 185 °C, respectively) compared to unmodified maple wood. The values of the R^2^ show a slight dependence speed of sound on temperature (R^2^ = 0.51 for SW; R^2^ = 0.49 for MW). The above-mentioned changes in the moduli of elasticity after thermal modification are not statistically significant (F = 0.93, *p* > 0.05) and the coefficient of determination was also low (for SW, it was 0.26 and for MW 0.49), respectively. This result corresponds with the results of Bekhta and Niemz [23]. Rusche [51] presents that irrespective of treatment and wood species, the decrease in MOE was significant when the mass loss exceeded 8%.

### 3.3. Effect of Thermal Modification on Acoustic Constant

The mean value of the acoustic constant of spruce wood after thermal modification increased by 4.4%, 6.5 and 4.2% and the acoustic constant of maple wood increased by 6.1, 6.8 and 3.0%. Stronger correlation was found for spruce wood (R^2^ = 0.78) than for maple wood (R^2^ = 0.56). The effect of thermal modification on the acoustic constant was significant (F = 3.68, *p <* 0.05). The sound radiation coefficient describes how much the vibration of a body is damped due to sound radiation. Particularly in the case of idiophones, such as xylophones and soundboards, a large sound radiation coefficient of the material is desirable [27,29]. An increase in acoustic constant due to thermal modification of spruce is, therefore, welcome because soundboards prefer values over 10 m^4^·kg^−1^·s^−1^ [27,29]. In terms of the use on the back plate string instruments, this change is negative. On the other hand, this increase is welcome, for example, in making bassoons and others woodwind instruments and drums kits as they are favored for their bright sound. The acoustic constant for these instruments can be in the range 4 to 8 m^4^ ·kg^−1^ ·s^−1^ [52].

### 3.4. Effect of Thermal Modification on Speed of Sound

The speed of sound in a material depends on how quickly vibrational energy can be transferred through the medium. Therefore, the speed of sound in a material depends on the material and on the state of the material. A high speed of sound (over 5000 m·s^−1^) is preferred for soundboards [27,52]. Our results showed that the sound waves move faster in thermally modified wood species. The speed of sound has increased after the thermal modification in both wood species, although the density slightly has decreased. Speed of sound of spruce wood increased (0.8 to 1.8%) and ranged from 5589 to 5608 m·s^−1^, i.e., this modification is appropriate for the soundboards of string musical instruments [53]. A higher increase was found for maple (1.9 to 3.8%). These results are consistent with an increase in the modulus of elasticity after thermal modification. The similar results were published by Puszynski and Warda [7], Pfriem [8] as well as by Esteves and Pereira [54]. Positive correlation between the temperature of modification and speed of sound was observed for spruce (R^2^ = 0.57) and for maple wood (R^2^ = 0.52). These changes are not statistically significant (F = 2.14, *p* > 0.05).

### 3.5. Effect of Thermal Modification on Specific Modulus of Elasticity

Specific modulus of elasticity *E*_sp_ represents the stiffness to mass ratio and is also known as specific stiffness. It is one of the most important parameters of resonant wood. *E*_sp_ is directly proportional to the cell-wall substances [55]. Hardwoods, in contrast to softwoods, include a diverse range of cell types, e.g., fibers, vessels, and parenchyma [56]. Softwood lignin differs from hardwood lignin in its structural structure and content. Lignin content is higher in softwood (25 to 36%); hardwood contains less lignin (15 to 25%). Lignin gives wood strength by joining individual fibers into a compact whole. Thermal modification even at lower temperatures (below 190 °C) causes degradation of cellulose and hemicelluloses. As a result, the lignin content increases, increases mechanical strength and, conversely, decreases permeability. Our results confirm this. Specific modulus of elasticity *E*_sp_ of spruce increased by 6.0%, 6.8% and 3.0% and *E*_sp_ of maple increased by 7.4%, 7.8% and 7.8% (at 135 °C, 160 °C and 185 °C, respectively) compared to unmodified wood. Lower density gives a higher modulus of elasticity. Lower density also gives a higher *E*_sp_ [7], which means a positive impact on most of the acoustic properties of wood.

### 3.6. Effect of Thermal Modification on Logarithmic Damping Decrement

The logarithmic damping decrement (*ϑ*) is another important acoustic characteristic, and it is preferred to be small in wood with excellent acoustic quality [57]. Lower logarithmic damping decrements result in higher material acoustic values [58]. This characteristic (ϑ) is related to internal friction, and it cannot be correlated to the mass of wood or the volume of wood, but it closely correlates to their structure.

The logarithmic damping decrement (*ϑ*) was varied depending on the temperature of modification. The lowest reduction in logarithmic damping decrement (16%) was observed after modification at 185 °C in the case of spruce wood (*ϑ* drops 8.3%). The higher effect of thermal modification was observed when spruce wood was modified at temperatures 130 °C (ϑ drops 16.3%) and 160 °C (ϑ drops 13.8%). The lowest reduction in damping of maple wood was recorded at 160 °C (*ϑ* drops 3.6%) and the highest drops *ϑ* (=7.3%) at 185 °C. R^2^ values in the case of ϑ were (R^2^ = 0.51 for SW and R^2^ = 0.83 for MW). Duncan’s test showed that the changes in logarithmic damping decrement caused by the thermal modification (of both species) are not statistically significant (F = 0.91, *p* > 0.05). Our results showed a high dependence between *E*_sp_ and *ϑ* (R^2^ = 0.97 for SW and R^2^ = 0.69 for MW).

The vibrational properties (*E*_sp_ and logarithmic damping decrement ϑ) depend on the cell-wall structure. The transformation of lignin and the increase in the concentration contributes to the reduced water adsorption (lower equilibrium moisture content). The lower equilibrium moisture and, hence, reduced swelling and shrinkage, is characteristic of the thermally modified wood, i.e., the microfibrils’ angle in the S2 layer is smaller. The orientation of cellulose microfibrils in the S2 layer affects these properties the most. While *E*_sp_ is decreasing, damping ϑ is increasing, with an increasing microfibril angle. The microfibrils’ angle in S2 and the deflection of the fibers show significant correlations with the vibrational properties in the longitudinal direction but not perpendicular to the wood grains. The collective effect of the microfibril angle and grain angle on both *E_sp_* and *ϑ* is higher than their individual effects on these properties [59,60,61].

### 3.7. Effect of Thermal Modification on Loss Coefficient and Characteristic Acoustic Impedance

A measured logarithmic damping decrement was used to calculate the internal friction-loss coefficient (Equation (6)). Loss coefficient is important because it shows the amount of absorbed acoustic energy. It increases with higher moisture content, and it is reduced by thermal treatment [51,62]. Our results are consistent with this claim. The loss coefficient of spruce wood decreased by 17.4% (at 135 °C), 14.8% (at 160 °C) and 9% (at 185 °C); for maple, the decrease was lower at each modification temperature, i.e., 5.7%, 4% and 7.6%. Since low damping is required for the top plate (spruce) and higher damping for the back plate (maple), it can be stated that the heat treatment seems to be suitable in this aspect.

The characteristic acoustic impedance z of spruce wood is very similar to maple (around 2300 MPa∙s∙m^−1^ for SW and 2550 MPa∙s∙m^−1^ for MW). Due to thermal modification, z increased but not significantly. The lowest increase was recorded at 185 °C for both types of wood (around 1%). A low z value is required for the top plate of string instruments (between 1200 and 3392 MPa∙s∙m^−1^) and high value z (between 1680 and 5760 MPa∙s∙m^−1^) is required for percussion and woodwind instruments [29,63] because the sound should decay slowly.

### 3.8. Effect of Thermal Modification on Acoustic Conversion Efficiency (ACE) and on Sound Quality Factor Q

The acoustic conversion efficiency (*ACE*) reflects the ability of the material to convert vibration energy into sound energy [55,64]. *ACE* combines internal friction and sound radiation together. The highest *ACE* value was measured after thermal modification at 135 °C for both wood species (*ACE*_SW_ = 1401 m^4^∙kg^−1^∙s^−1^; *ACE*_MW_ = 438 m^4^∙kg^−1^∙s^−1^). Similarly, *Q*, as a wood sound quality descriptor, was highest at 135 °C (*Q*_SW_ = 105.3 and *Q*_MW_ = 60, 61). The *ACE* increase compared to untreated wood was 8.3% for SW and 11% for MW. The increase in *Q* of SW was considerably higher, up to 21%, while at MW, it was 7%. A high *ACE* value of spruce wood is especially required for soundboards because such a board should show the maximum ratio of the sound radiation to internal friction, i.e., a high *ACE* and low loss coefficient *η* is preferred [54].

### 3.9. Effect of Thermal Modification on Timber of Sound

The FFT analysis demonstrated a positive effect of the thermal modification on the timbre of sound of the maple specimens, as well as of the spruce wood specimens (Figure 4 and Figure 5). Influence of thermal modification on the acoustic constant as well as on the frequency spectrum (FFT analysis), which both investigated the wood species, were identical. The higher value of the acoustic constant is characterized by the increasing of acoustic energy radiation, as confirmed by increasing the sound pressure levels (Figure 4 and Figure 5).

The results of the FFT analysis (time course of signal, sound spectrum) of the bars of spruce and maple wood before (blue curve) and after modification (red curve) are presented in Figure 4 and Figure 5. The mean ratios of aliquot frequencies in maple wood after thermal modification were as follows: 1:1.6:2.4 at 135 °C (1:1.7:2.6 before modification), 1:1.7:2.9 at 160 °C (1:1.8:2.8 before), 1:1.7:2.7 at 185 °C (1:1.6:2.6 before). Figure 4 shows that the fundamental resonant frequency of maple slightly increased after each modification and the sound pressure level reduced the thermal modification effect.

The sound pressure level of low frequencies decreased and of higher frequencies increased, which means that the lower frequencies are repressed, and the higher frequencies are highlighted, so the timber of sound will be sharper [65].

## 4. Conclusions

As indicated by our experiments, we can improve the properties of wood for musical instruments by means of thermal modification at lower temperatures (135 °C to 185 °C). This modification causes a decrease in the mass and density but increase in the speed of sound and MOE, as well as the acoustic constant. These changes are not statistically significant, except for the acoustic constant.

Thermal modification leads to changes in vibrational properties by increasing the specific modulus of elasticity *E*_sp_ and decreasing the loss coefficient *η*. Our results showed a high dependence between *E*_sp_ and *ϑ*.

The characteristic acoustic impedance z of spruce and maple slightly increased.

The thermal modification process is the cause of not only the changes in the wood properties but also the changes in timbre and sound quality.

## Figures and Tables

**Figure 1 polymers-14-02813-f001:**
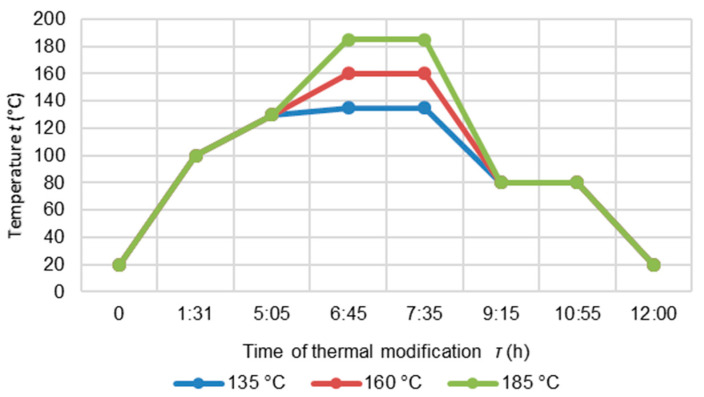
Diagram of the thermal modification process.

**Figure 2 polymers-14-02813-f002:**
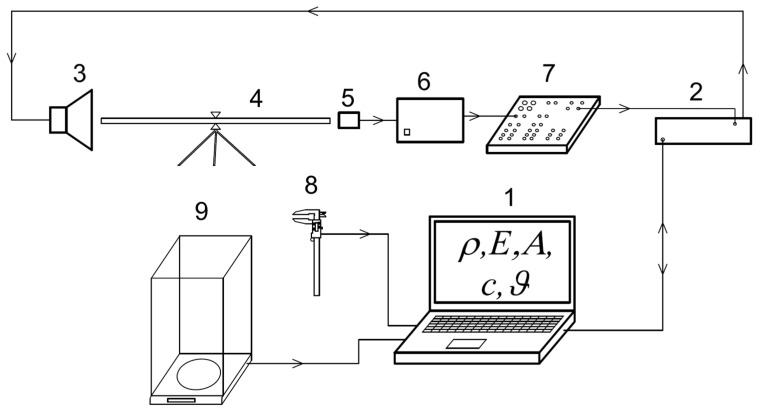
Measuring device MEARFA [36].

**Figure 3 polymers-14-02813-f003:**
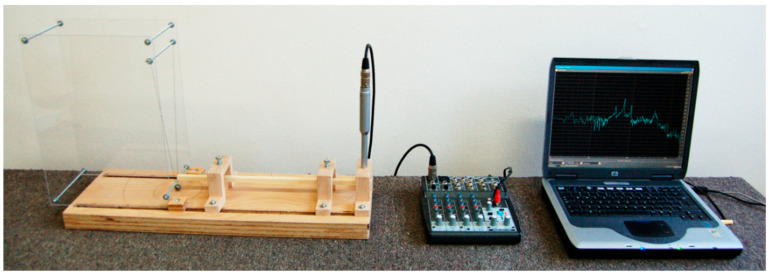
Experimental device for FFT analysis.

**Figure 4 polymers-14-02813-f004:**
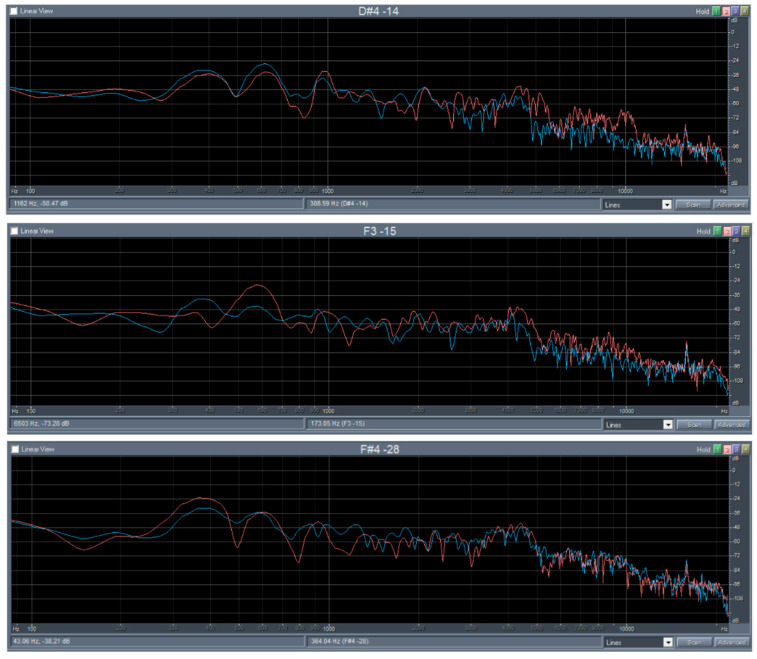
FFT analysis maple before and after thermal modification at 135 °C, 160 °C and 185 °C.

**Figure 5 polymers-14-02813-f005:**
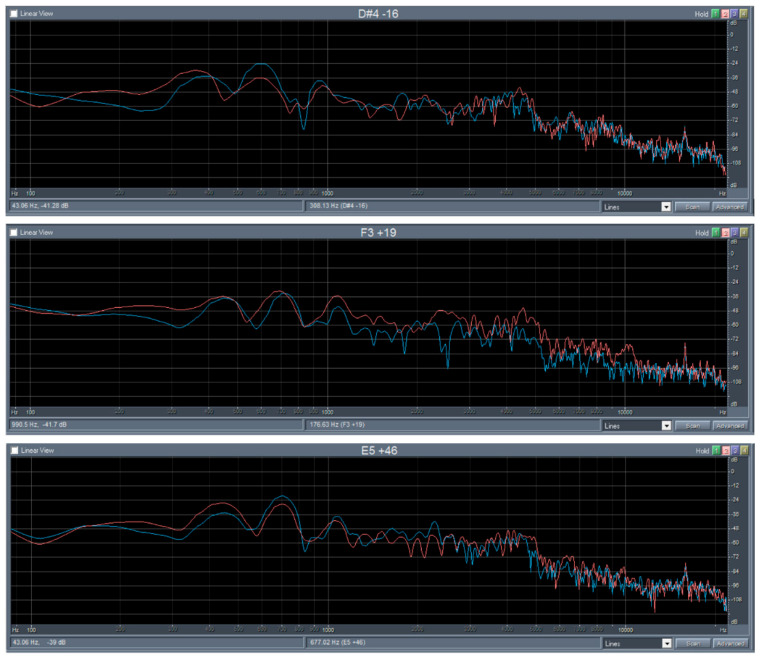
FFT analysis spruce before and after thermal modification at 135 °C, 160 °C and 185 °C.

**Table 1 polymers-14-02813-t001:** PACH (density *ρ*, dynamic modulus of elasticity *E*_L_, acoustic constant *A*, speed of sound *c*_L_, resonant frequency *f*r, logarithmic damping decrement *ϑ*, loss coefficient *η*, characteristic acoustic impedance *z*, acoustic conversion efficiency (*ACE*) and sound quality factor *Q)* of spruce and maple wood before and after thermal modification.

PACH		Norway Spruce				Sycamore Maple			
		Unmod.	135 °C	160 °C	185 °C	Unmod.	135 °C	160 °C	185 °C
*ρ* (kg∙m^−3^)	MV	425	420	413	415	609	601	603	604
	SD	32.8	21.6	13.8	17.9	38.6	24.2	19.6	20.5
	y = 0.0671x + 426.64 R^2^ = 0.8274					y = −0.037x + 608.87 R^2^ = 0.628			
*E*_L_ (GPa)	MV	12.48	13.12	12.99	12.63	10.70	11.35	11.43	11.02
	SD	0.025	0.018	0.020	0.017	0.034	0.021	0.028	0.0361
	y = 0.0021x + 12.544 R^2^ = 0.257					y = 0.0032x + 10.723 R^2^ = 0.4911			
*E*_sp_ (10^6^∙m^2^∙s^−2^)	MV	29.36	31.24	31.45	30.43	17.57	18.88	18.95	18.24
	SD	0.031	0.023	0.019	0.13	0.33	0.24	0.25	0.28
	y = 0.0098x + 29.396 R^2^ = 0.5668					y = 0.0064x + 17.614 R^2^ = 0.519			
*c*_L_ (m·s^−1^)	MV	5419	5589	5608	5517	4192	4346	4353	4271
	SD	390	228	263	281	285	223	231	197
	y = 0.089x + 5422 R^2^ = 0.5739					y = 0.747x + 4197 R^2^ = 0.5215			
*A* (m^4^∙kg^−1^·s^−1^)	MV	12.75	13.31	13.58	13.29	6.88	7.23	7.22	7.07
	SD	1.15	0.88	0.96	0.86	0.82	0.63	0.68	0.65
	y = 0.0042x + 12.707 R^2^ = 0.7767					y = 0.0017x + 6.89 R^2^ = 0.5587			
*f*_r_ (Hz)	MV	6025	6211	6238	6192	5240	5410	5382	5328
	SD	527	462	428	483	535	436	323	341
	y = 0.0042x + 12.105 R^2^ = 0.8241					y = 0.0042x + 12.105 R^2^ = 0.8241			
ϑ (-)	MV	0.036	0.030	0.031	0.033	0.055	0.052	0.053	0.051
	SD	0.00369	0.00312	0.00279	0.00264	0.00427	0.00381	0.00363	0.00356
	y = −3∙10^−5^ x+ 0.0358 R^2^ = 0.5142					y = −2∙10^−5^ + 0.0554 R^2^ = 0.8283			
*η* (-)	MV	0.0115	0.0095	0.0099	0.0105	0.0175	0.0165	0.0168	0.0162
	SD	0.0018	0.0009	0.0011	0.0012	0.0021	0.0019	0.0017	0.0019
	y = −9∙10^−6^x + 0.0114 R^2^ = 0.5108					y = −7∙10^−6^x + 0.0176 R^2^ = 0.8532			
*z* (MPa∙s∙m^−1^)	MV	2303	2347	2316	2290	2553	2612	2625	2580
	SD	12.84	4.79	3.63	5.03	11.01	5.35	4.53	4.39
	y = 0.0072x + 2313 R^2^ = 0.0005					y = 0.2966x + 2555 R^2^ = 0.4449			
*Q*	MV	86.96	105.26	101.01	95.23	56.82	60.61	59.52	61.73
	SD	5.8	4.1	3.7	3.9	5.2	4.6	4.2	3.8
	y = 0.0734x + 87.938 R^2^ = 0.4568					y = 0.0268x + 56.324 R^2^ = 0.8613			
*ACE* (m^4^∙kg^−1^·s^−1^)	MV	1109	1401	1372	1266	393.1	438.2	429.8	436.4
	SD	183	165	162	157	108	98	96	93
	y = 1.3508x + 1118 R^2^ = 0.556					y = 0.2714x + 390.45 R^2^ = 0.8767			

## Data Availability

Not applicable.

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
