# Peer review of "Thermal Modification of Spruce and Maple Wood for Special Wood Products"

_polymers, 2022, doi:10.3390/polym14142813_

Round 1
Reviewer 1 Report
Well designed study that is properly presented. I have no specific scientific points that require improvement. I recommend to change lay out of fig. 4 and 5 to improve readability. I suggest a white instead of black background.
Author Response
Response Letter to Reviewer 1
Dear Reviewer 1:
thank you for your kind review of our article and recommendations. We are very sorry, but we cannot accept your proposal to change the background of Fig. 4 and 5. The presented Fig. 4 and 5 are the output from the used software.
Once again, thank you very much for your review and positive comment.
Yours faithfully,
Anna Danihelová, Zuzana Vidholdová, Tomáš Gergeľ, Lucia Spišiaková Kružlicová and Michal Pástor
Reviewer 2 Report
The presented manuscript is focused on investigating selected physical and acoustic characteristics of thermally-modified spruce and maple wood, intended for manufacturing special wood products, mainly musical instruments. In general, the manuscript is very well-written, structured and informative, and can be accepted for publication in the Polymers Journal in its current form.
I have only some minor comments on your work:
Overall, the title, abstract and the keywords correspond to the aims and objectives of the manuscript.
The abstract of the manuscript (lines 15 to 28) and the keywords (lines 29-30) correspond to the aims and objectives of the paper. The abstract is informative, and contains the main findings of the article.
Line 24: please use only the abbreviation “PACHs” for the physical and acoustic characteristics, it has already been introduced in line 17.
Line 58: please replace “doesn’t” with “does not”.
In general, the introduction section is very well-prepared, structured and informative, and the aim of the research is clearly outlined and justified.
Lines 103 – 104: please use Italics for the botanical names of the tree species, i.e. Picea abies (L.) Karst and Acer pseudoplatanus L.
Line 106: please write the dimensions of test specimens as 10 mm x 10 mm x 400 mm.
Line 226: “…by hitting is at the bottom…” should be “….by hitting it at the bottom…”, a typo mistake, please revise.
Overall, the Materials and Methods sections is clear, descriptive and very well presented in the necessary details.
Lines 245-246: please move the information about the statistical software to the previous section of the manuscript.
The results obtained from the study are clearly presented, detailed, and properly discussed with relevant research works.
The Conclusion part reflects the main findings of the manuscript.
The References cited are appropriate to the research topic. Some of them are not formatted according to the Journal requirements. Please refer to the Instructions for Authors.
Best regards!
Author Response
Response Letter to Reviewer 2
Dear Reviewer 2,
thank you for your kind review of our article, comments and recommendations. We accept your recommendations to editing the text.
The correction text of article according to your recommendations.
I have only some minor comments on your work:
Overall, the title, abstract and the keywords correspond to the aims and objectives of the manuscript.
The abstract of the manuscript (lines 15 to 28) and the keywords (lines 29-30) correspond to the aims and objectives of the paper. The abstract is informative, and contains the main findings of the article.
Line 24: please use only the abbreviation “PACHs” for the physical and acoustic characteristics, it has already been introduced in line 17.
Thank you for your recommendation. We used the abbreviation “PACHs” in line 24.
Line 58: please replace “doesn’t” with “does not”.
We replaced “doesn’t” with “does not” in line 58.
In general, the introduction section is very well-prepared, structured and informative, and the aim of the research is clearly outlined and justified.
Lines 103 – 104: please use Italics for the botanical names of the tree species, i.e. Picea abies (L.) Karst and Acer pseudoplatanus L.
We did not notice this error while writing the text. Thanks for this suggestion to edit the Latin botanical tree names. We have used italics for the Latin botanical names of tree species.
Line 106: please write the dimensions of test specimens as 10 mm x 10 mm x 400 mm.
We wrote the dimensions of test specimens as 10 mm x 10 mm x 400 mm according to reviewer's recommendation.
Line 226: “…by hitting is at the bottom…” should be “….by hitting it at the bottom…”, a typo mistake, please revise.
We have corrected this typo mistake.
Overall, the Materials and Methods sections is clear, descriptive and very well presented in the necessary details.
Lines 245-246: please move the information about the statistical software to the previous section of the manuscript.
The information about the statistical software were moved to section "2. Material and Methods".
The results obtained from the study are clearly presented, detailed, and properly discussed with relevant research works.
The Conclusion part reflects the main findings of the manuscript.
The References cited are appropriate to the research topic. Some of them are not formatted according to the Journal requirements. Please refer to the Instructions for Authors.
The cited References were formatted according to the Journal requirements.
Once again, thank you very much for your review and positive comment.
Yours faithfully,
Anna Danihelová, Zuzana Vidholdová, Tomáš Gergeľ, Lucia Spišiaková Kružlicová and Michal Pástor.